# Cerebrolysin in Patients with TBI: Systematic Review and Meta-Analysis

**DOI:** 10.3390/brainsci13030507

**Published:** 2023-03-17

**Authors:** Konrad Jarosz, Klaudyna Kojder, Agata Andrzejewska, Joanna Solek-Pastuszka, Anna Jurczak

**Affiliations:** 1Department of Specialist Nursing, Pomeranian Medical University, Rybacka 1, 70-204 Szczecin, Poland; 2Anesthesiology and Intensive Care Department, Pomeranian Medical University, Rybacka 1, 70-204 Szczecin, Poland; 3Anesthesiology and Intensive Care Department, University Hospital 1, Unii Lubelskiej 1, 72-252 Szczecin, Poland

**Keywords:** TBI, Cerebrolysin, neuroprotective treatment

## Abstract

TBI (traumatic brain injury) is one of the most common causes of deaths and failure to return to society according to the latest statistics. Cerebrolysin is a drug approved for use in patients diagnosed with TBI. It is a mixture of neuropeptides derived from purified porcine brain proteins and multiple experimental studies have proven its neuroprotective and neurorestorative properties both in vitro and in vivo. In our meta-analysis, we analyze the latest clinical study reports on the use of Cerebrolysin in patients with TBI. The authors searched the databases: Pub Med, Cinahl, Web Of Science, and Embase from database inception until 11th July 2022. Ten clinical studies were eligible and included in the final analysis, including both retrospective and prospective studies of 8749 patients. Treatment with Cerebrolysin was associated with a statistically significant change in GCS and GOS. Mortality of any cause and the length of stay was not affected by the treatment. Our findings support and confirm the beneficial effects of Cerebrolysin treatment on the clinical outcome of patients after TBI. Further multi-center studies to optimize dosing and time of administration should be conducted.

## 1. Introduction

TBI is one of the most common causes of death and failure to return to society according to recent literature. Depending on the region, the mortality rate due to TBI ranges from 13/100,000 (China) [1] to 11/100,000 (Europe) and 17/100,000 (USA) [2]. Attempts to implement treatment algorithms based on the Lund concept through the first Brain Trauma Foundation (BTF) recommendations led to a reduction of mortality among patients diagnosed with TBI [3,4]. BTF established first guidelines with American Association of Neurological Surgeons (AANS), Congress of Neurological Surgeons (CNS), and AANS/CNS Joint Section on Neurotrauma and Critical Care in 2007 with an update in 2016. The algorithms are based on the literature review. The main goals of the therapy are to keep the ICP below 22 mmHg and maintain Systolic Blood Pressure at particular levels for different ages. For CPP the main target should be 60–70 mmHg according to those guidelines. It changed the approach for TBI treatment from the Lund concept, which included, among others, albumin and blood transfusion. The most recent approach is represented by the Seattle International Severe Traumatic Brain Injury Consensus Conference (SIBICC). In the construction of these assumptions, the Delphi method was used. A group of experts from around the world were questioned about the therapeutic and diagnostic consensus concerning TBI treatment and monitoring. Based on their opinions, algorithms were proposed, in which the appropriate treatment depends on monitoring results. Although the treatment results have been improving in the last 30 years, mortality still remains high worldwide in patients diagnosed with TBI. In the diagnosis and treatment of TBI, the main emphasis is now to limit the damage involved in secondary trauma. Diagnosis and treatment of secondary ischemia, including rupture of the blood–brain barrier, edema, and, as a consequence, hypoxia, is currently the focus of most efforts and algorithms [3,5]. However, there is no consensus on procognitive, neuroprotective treatment, both in terms of the medication used and the time and dosage of their administration.

Cerebrolysin is a low molecular weight neuropeptide preparation obtained from purified porcine brain proteins that has proven neuroprotective properties in vitro and in vivo, including modeling the permeability of endothelial membranes and anti-inflammatory effects [6,7,8]. Fiani et al. distinguished the main activities of Cerebrolysin according to periods in the development of the pathology. Close-to-injury Cerebrolysin acts in neuroprotective way by influencing the neuroinflammation (reducing free radicals and proapoptotic factors). In TBI, the neurotrophic activity is very important, where Cerebrolysin by similarity to NTF activity activates the PI3K pathway. In the second stage, the main directions of acting are regeneration and neuroplasticity. On that level, Sonic hedgehog (Shh) signaling is worth paying attention to, as well as the Gli complex that is activated via Shh. With the Shh influence, Cerebrolysin acts also on neurogenesis and gliogenesis (especially oligendendrogenesis). In the further period, Cerebrolysin enhances neuroplasticity by increasing synaptic density and preserving the neural communication [9]. Cerebrolysin is considered as a co-treatment in several medical acute and chronic conditions. Because of its neurogenesis stimulative, neuronal repair, and neuroprotective properties, it has been administered in patients with acute traumatic brain injury, stroke, subarachnoidal hemorrhage diagnosis, and also with slower progressing neurodegenerative diseases such as Parkinson’s disease, Alzheimer disease, or multiple sclerosis [6]. In their work, Zhang et al. investigated the influence of Cerebrolysin on treatment results in Wistar rats. They concluded that Cerebrolysin changes the number of neurons in the region of dentate gyrus and the hippocampus. It also protects the main dendrite integrity in the striatum region [6]. Those findings confirm that Cerebrolysin may have a beneficial influence in the processes of memory and learning, which is crucial after recovering from trauma. Cerebrolysin in comparison with saline had a positive influence on long-term spatial learning and non-spatial memory in rats after experimental closed mild traumatic brain injury [7]. Cerebrolysin acts similar to Neurotrophic Factor (NTF) in order to save the brain tissue from exacerbated inflammatory response [6,9]. Although the secondary trauma of TBI will always lead to neuroinflammation, prolonged uncontrolled inflammation will always lead to neuronal loss and degeneration. The NTF family comprise small proteins or peptides that include at least four groups of factors responsible for immunological homeostasis. We distinguish Neurotrophins, the Cilliary Neurotrophic Factor subgroup, Glial-cell line-derived neurotrophic factor subgroup, Ephrins and Epidermal Groth Factor, and Transforming Growth Factor subgroup. The other mechanism of Cerebrolysin neuroprotection involves Sonic hedgehog (SHH) pathways. Shh is a part of the Hh hedgehog family. It plays a significant role in embryonic but also adult neuronal damage signaling. Recent TBI models showed that the Shh pathway is exacerbated after cortical injury [6,9,10]. Cerebrolysin modulates mRNA expression in order to activate the Shh pathway itself but also by Shh receptors activation (gliogenesis, neurogenesis) [9,10,11]. The effector for Shh pathway signaling is the Gli protein complex. The Gli complex is involved not only in developmental processes but also neurorecovery in pathological conditions. Acting on neurotropic agents Cerebrolysin changes the activity of GABAergic and cholinergic pathways in the brain. Cerebrolysin also has been proven to affect particularly on microglia cells. Those glia cells control the homeostasis and immunological responses, and Cerebrolysin modulates their function in order to limit the exacerbated inflammation response [12].

In vivo studies with rats have shown the reduction of astrocyte activation, reduction in axonal damage, and increase in neurogenesis in closed experimental brain injury [13]. In vivo studies have with patients also shown its beneficial role in slowing down of EEG activity and improving rehabilitation and results in terms of cognitive performance [14,15].

According to the recommendations, Cerebrolysin could be administrated in various dosages at 10–50 mL per day. The intravenous infusion should be prepared in 100 mL total volume after adding the Cerebrolysin to 0.9% saline or Ringers or 5% glucose solution. The duration of infusion should take at least 15 min.

The time of administration seems to be crucial, especially in the condition of TBI. In animal studies, it has been shown that early administration is associated with better outcomes, in terms of sensory–motor functions, brain edema, and blood–brain barrier leakage [16].

Contraindications for Cerebrolysin administration are: allergy, seizure, severe renal failure. 

The aim of the following meta-analysis was to search the literature in terms of the analysis of the clinical effect of Cerebrolysin on the mortality, Glasgow Outcome Scale (GOS), Glasgow Coma Scale (GCS), and Length Of (Hospital) Stay (LOS) in patients after traumatic brain injuries.

## 2. Materials and Methods

### 2.1. Search Strategy and Inclusion Criteria

Two independent authors (KK and KJ) searched the Pub Med, Cinahl, Web Of Science, and Embase databases from the database’s inception until 11th July 2022 with language restriction (only English) for studies aiming to evaluate the effect of Cerebrolysin in the treatment effect of patients with TBI diagnosis. The search strings the authors used are listed in Table 1 below.

The electronic search was supplemented by a manual review of the reference lists from eligible publications and relevant reviews.

The following inclusion criteria were applied:Human studiesAdult patients (>18 y)Diagnosis of mild, moderate, or severe TBI (head trauma, brain trauma)

Exclusion criteria were as follows:Animal studies, in vitro studies, reviews, systematic reviews, editorials, individual case reports, and opinion, editorial, or perspectives articlesPediatric patients (<18 y)Pregnant patientsMultiorgan failureStudies in language other than English

#### 2.1.1. Data Abstraction

Data on study design, patient characteristics, and treatment (dosage of Cerebrolysin, trial duration) were independently extracted from each study in accordance with the Preferred Reporting Items for Systematic Reviews and Meta-Analyses (PRISMA) standard by two independent investigators (KJ and KK). Whenever data were missing for the review, authors were contacted for additional information. Inconsistencies were resolved by consensus with a senior author (JSP).

#### 2.1.2. Outcomes

Co-primary outcomes were the GOS and GCS, as well as mortality and LOS, all reflecting drug efficacy in patients admitted to hospital due to TBI.

#### 2.1.3. Data Synthesis and Statistical Analysis

We conducted a random effects meta-analysis of outcomes, in which ≥2 studies contributed data, using Comprehensive Meta-Analysis V3 (http://www.meta-analysis.com (accessed on 28 August 2022) [17]. Study heterogeneity was determined using the chi-square test of homogeneity, with *p* < 0.05 indicating significant heterogeneity. All analyses were two-tailed with alpha equal to 0.05.

For continuous outcomes, we analyzed the standardized differences in means of endpoint scores using observed cases (OC). Categorical outcomes were analyzed by risk ratio (RR). We aimed to conduct subgroup and exploratory maximum likelihood random effects meta-regression analyses of the co-primary outcomes (e.g., with age, sex), however, due to insufficient data, we were not able to conduct this. Finally, we inspected funnel plots and used Egger’s regression test as well as the Duval and Tweedie’s trim and fill method to quantify whether publication bias could have influenced the results [18,19].

## 3. Results

### 3.1. Search Results

The initial search yielded 27 results. There were four studies, which were excluded being duplicates after evaluation on the title or abstract. There were no additional articles identified via a hand search. Then, 23 full-text articles were reviewed. Of those, 13 were excluded due to not fitting inclusion criteria. Reasons for exclusion were wrong study aim (*n* = 3), wrong language (*n* = 2), no access to full text (*n* = 4), duplicate (*n* = 3), and animal study (*n* = 1), yielding 10 studies, which were included in the meta-analysis. The flowchart of the database search is presented below in the Figure 1.

### 3.2. Study, Patients, and Treatment Characteristics

Altogether, 10 studies (*n* = 8749) were included and are presented in Table 2.

Three studies were designed as blinded. The intervention was administered to patients for 5–30 days, with a total study duration of up to 6 months. In all studies, patients were given intravenously Cerebrolysin at a dosage of either 10 mL/day, 20 mL/day, 30 mL/day, or 50 mL/day i.v. A total of 10 mL/day was the dosage for Khalil et al. and Ashgari et al. [20,21]. A total of 20 mL/day was the dosage for Muresanu et al., 2015 [22]. The most common dosage was 30 mL/day, administrated in the work of Alvarez et al. in both 2003 and 2008 and by Chen et al., Lucena et al., and Muresanu et al. in 2015 [22,23,24,25,26]. A total of 50 mL/day was administrated Poon et al., Wong et al., and Muresanu et al. in 2020 [14,15,27]. Additionally, the treatment duration time was different in presented studies. The time varied from 5 days (Chen et al.), 10 days (Muresanu et al., 2015), 20 days (Poon et al., Wong et al., Muresanu et al., 2020), and 20–30 days (Alvarez et al., 2003, Alvarez et al., 2008, Lucena et al., Khalil et al.). Only a few patients demonstrated seizures during hospital stay, as reported by the research by Alvarez et al., Chen et al., and Khalil et al. [20,24,25]. Other authors did not attach the information including seizure. Some of the patients were subjected to craniotomy, as described in Table 3 below. In addition, in Table 4 below, the treatment initiation time, initial GCS, TBI severity in Cerebrolysin, and control groups in analyzed papers are presented. As we can see, the treatment administration varies from 24 h to >20 months. The main age varied from 30.1 to 64 years. In all analyzed studies, the percentage of men was higher than the percentage of women.

### 3.3. Effects on GCS Score at Endpoint

Using random-effects analysis, the DM for the GCS score at endpoint in patients treated with Cerebrolysin compared to control arms was 1.344 with a 95% confidence interval of −0.258 to 2.945; *p* = 0.1. The heterogeneity was high (Q value = 34.458; df =2; *p* = 0.0; I2 = 94.196). Results are shown in Figure 2. An Egger’s test did not suggest a publication bias regarding the net effect of Cerebrolysin on GCS score at endpoint (Egger’s test: *p* = 0.71).

### 3.4. Effects on GOS Score at Endpoint

Using random-effects analysis, the DM for the GOS score at endpoint in patients treated with Cerebrolysin, compared to non-interventional arms, was 0.422 with a 95% confidence interval of 0.262 to 0.581; *p* = 0.000. The heterogeneity was high (Q value = 20.377; df = 6; *p* = 0.002; I^2^ = 70.55). Results are presented in Figure 3 and Figure 4.

An Egger’s test did not suggest a publication bias regarding the net effect of Cerebrolysin on GOS (Egger’s test: *p* = 0.14).

### 3.5. Effects on Hospital LOS

Using random-effects analysis, the DM for the LOS in patients treated with Cerebrolysin compared to non-interventional arms was −1.255 with a 95% confidence interval of −6.422 to 3.913; *p* = 0.634. The heterogeneity was high (Q value = 20.182; df = 3; *p* = 0.0; I2 = 85.135). Results are presented in Figure 5 and Figure 6.

An Egger’s test did not suggest a publication bias regarding the net effect of Cerebrolysin on LOS (*p* = 0.76).

### 3.6. Effects on Mortality

Using random-effects analysis, the risk ratio for mortality in patients treated with Cerebrolysin compared to non-interventional arms was 0.469 with a 95% confidence interval of 1.174 to −1.596. The heterogeneity was low (Q value = 1.555; df = 2; *p* = 0.46; I2 = 0). Results are presented in Figure 7 and Figure 8.

An Egger’s test did not suggest a publication bias regarding the net effect of Cerebrolysin on mortality (*p* = 0.51).

No other variables were analyzed due to insufficient data. However abstracted data is placed in Table 1.

## 4. Discussion

Our meta-analysis compares the treatment result among the clinical studies found in Pub Med, Cinahl, Web Of Science, and Embase from database inception until 11th July 2022 describing the effect of Cerebrolysin on patients diagnosed with TBI. The studies included both prospective randomized studies [15,25,27] and observational or historical cohorts [20,21,22,23,26]. In most of the analyzed studies, the authors point out the heterogeneity of the groups and the lack of consensus regarding the dose of the drug and the duration of its use [15,27]. Most studies have proven that Cerebrolysin has a positive treatment effect in TBI patients in terms of cognitive functions, GOS, and GCS, but does not alter the mortality rate or LOS [21,22,28,29].

The limits of this meta-analysis include the lack of large, randomized trials, different doses of the drug administered at different times after primary trauma, heterogeneous group of respondents, and heterogeneous results in terms of outcome. Therefore, it was not possible to compare all the publications mentioned in all planned aspects.

The heterogeneity of the group’s results from the diversity of individual publications regarding the inclusion and exclusion criteria, the severity of injury, the type of study, and the study methodology. Concerning the severity of injury, most of the patients were diagnosed with a moderate—severe TBI. The drug dose ranged from 10 mL in Ashgari et al. to 50 mL in Muresanu et al., 2020 and Poon et al. [15,21,27]. The initiation of Cerebrolysin treatment varied from 24 h to >20 months.

In a more basic approach, however, the diversity resulted mainly from the heterogeneity of the pathology itself. TBI is diagnosed in an incredibly heterogeneous group of patients in terms of both the severity of the injury, the study population, and in the method of basic treatment. The implementation of guidelines from the Brain Trauma Foundation or the Seattle consensus, or managing patients with other types of monitoring algorithms, e.g., CT alone, was not common in most of the responders [3,5,30]. In the latest registries of TBI, ICP monitoring covered less than 50% of monitored patients in Europe and in China [31,32]. Those registries, based on living systematic reviews partially as part of as part of the Collaborative European NeuroTrauma Effectiveness Research in Traumatic Brain Injury (CENTER-TBI) project, were designed to assess all information concerning TBI epidemiology including age, sex, mortality, and severity of trauma. We can only assume that the rest of the patients were treated according to even more heterogenous algorithms. The observed treatment effect of Cerebrolysin or any other treatment would, therefore, depend not only the intervention itself but also on many other confounding factors, which explains the differences in the results and the heterogeneity of groups compared in research.

The positive effect of Cerebrolysin has been proven in vitro (decrease in microglial activity, excitotoxicity, production of free radicals, increase in neuronal survival) and has also been demonstrated in vivo in animal studies [7,16,33,34], as well as in clinical studies [15,20,21,22,23,24,25,26,27,28]. In particular, the role of modeling the immune response deserves attention. Acting through the NTF, as well as influencing the activity of GABAergic and cholinergic pathways Cerebrolysin, affects the response to the primary injury and could have an influence on the secondary injury in the damaged brain. It is worth noticing that Cerebrolysin could moderate the processes at all stages after the initial trauma because of its neuroprotective properties. That also means that it could be beneficial after introducing the treatment at every level after the initial TBI—in the first 24 h, but also after 20 months since the injury. The primary work of Sharma and Zhang and Chopp concluded that the effect of Cerebrolysin treatment is dose- and time-dependent in animal studies, and it is reasonable to assume that the clinical effect in human studies will also depend on the dose and time of administration [7,16]. Of note, positive treatment effects of Cerebrolysin in TBI are also evident in a rehabilitation setting after delayed administration of Cerebrolysin long after the actual injury [23,24].

In analyzed research, the authors mentioned different outcomes measured as main findings of their work. In the work of Muresanu et al., the 2015 work’s results showed that Cerebrolysin in dosages 20 mg/day and also 30 mg/day improved the GOS and RDS (Modified Rankin Disability Score) in moderate and severe patients after 10 and 30 days after TBI [22]. The results of Alvarez et al. from 2003 and 2008 indicate that Cerebrolysin might encourage the EEG-activation effect in the post-acute period after trauma in moderate to severe TBI patients [23,24]. Chen et al. in 2013 showed that Cerebrolysin improves the Cognitive Abilities Screening Instrument (CASI) score in patients with mild TBI [25]. Khalil et al. in 2017 stated that Cerebrolysin treatment in patients with severe TBI is associated with increased Glasgow Outcome Scale Extended (GOSE) score and reduced mortality [20]. Lucena et al. in 2022 in their work with severe TBI patients confirmed that Cerebrolysin has a beneficial influence on GCS, GOS, and LOS [26]. Ashgari et al. in 2014 stated that GCS was significantly higher in treatment groups in comparison with control [21]. The CAPTAIN I and II trials according to Poon et al. in 2019 and Muresanu et al. in 2020 elaborated on the cognitive improvement of patients (Stroop test and Color Trail Test) with additional treatment of Cerebrolysin [15,27].

To this date, several manuscripts summarizing the effects of Cerebrolysin and other neuroprotective drugs have been published [29,35,36]. One meta-analysis involving Cerebrolysin in TBI by Ghaffarpasand et al. in 2018 has been performed in [28]. The authors included the investigation of Wong et al., 2005, Alvarez et al., 2008, Ashgari et al., 2014, Muresanu et al., 2015, and Khalil et al., 2017. In our work, we confirmed some of their results after adding the work of: Alvarez et al., 2003, Chen et al., 2012, Poon et al., 2019, Muresanu et al., 2020, and Lucena et al., 2022. The main conclusion of Ghaffarpasand et al. was that the Cerebrolysin in TBI treatment improved mRS and GOS. Our research confirmed the improvement in GOS in patients with TBI diagnosis treated with Cerebrolysin

In the basic treatment of TBI, we generally followed the Traumatic Brain Foundation algorithms [3]. In the case of ICP-monitored patients, we can also follow the SIBICC or other algorithms when we do not have the possibility to estimate the ICP in a direct, invasive way (among others, algorithms following the changes in CT imaging) [3,4,5]. However, even in the protocol of TBF, it is stated that the level of evidence for the interventions hardly exceeds Level II. Furthermore, in the situation of a very heterogenous disease, there is great difficulty in designing and performing prospective randomized trials that would provide conclusive answers regarding the optimal dosage and time of administration for Cerebrolysin or any other intervention. Under such circumstances, it is worth paying attention to algorithms created on the basis of the consensus of expert panels, such as the SIBICC or Guidelines for Cognitive Rehabilitation Following Traumatic Brain Injury, 2023, where Cerebrolysin is mentioned as a compound worth considering in the treatment of TBI [5,37]. The INCOG guidelines represent the consensus made by a panel of experts based on the publications searched from 2014. It includes pharmacological and non-pharmacological strategies to improve attention after moderate to severe TBI. As the post-TBI period is associated with disorders of neurotransmitters affecting cognitive abilities and attention—this paper proposes among others pharmacological substances that positively affect the improvement of attention. According to the authors, Cerebrolysin is one of the drugs that can be used with potential benefit in this group of patients. The authors point out, however, that there is a lack of broader evidence from the studies at present [37].

## 5. Conclusions

Our meta-analysis confirms the positive treatment effect of Cerebrolysin on clinical outcomes in the aspects of GCS and GOS in patients with TBI. We were not able to detect significant effects on mortality or LOS, possibly also due to the relatively small sample size. Therefore, more randomized studies in the field of Cerebrolysin in TBI are needed to confirm and extend our findings with respect to the clinical utility of Cerebrolysin in TBI.

## Figures and Tables

**Figure 1 brainsci-13-00507-f001:**
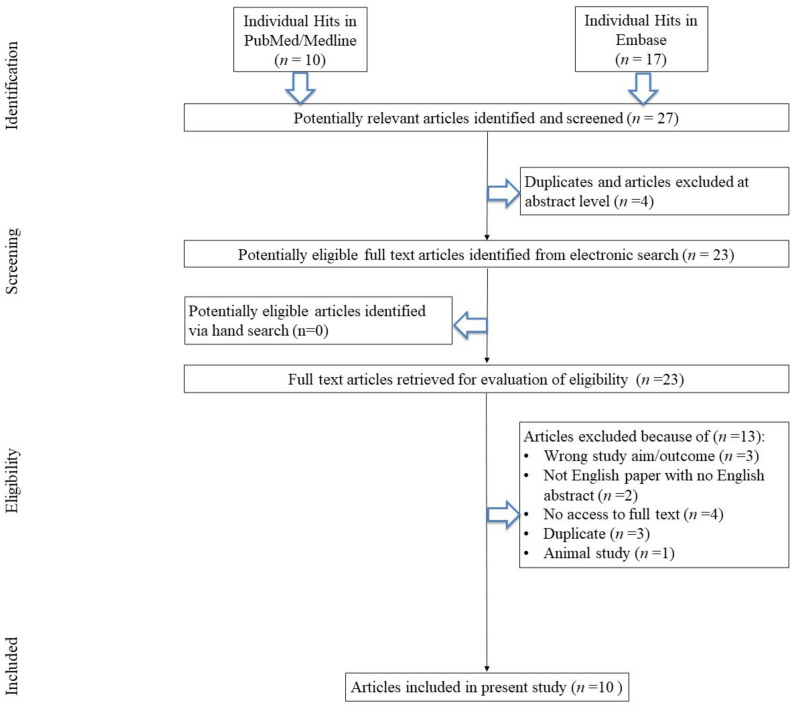
Study flow chart.

**Figure 2 brainsci-13-00507-f002:**
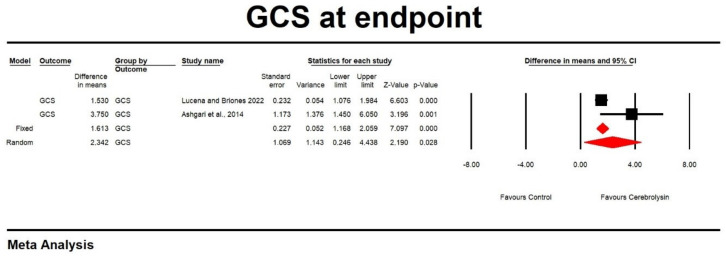
Effect of Cerebrolysin on GCS score (Z value = 2.190, *p* = 0.028). [21,26].

**Figure 3 brainsci-13-00507-f003:**
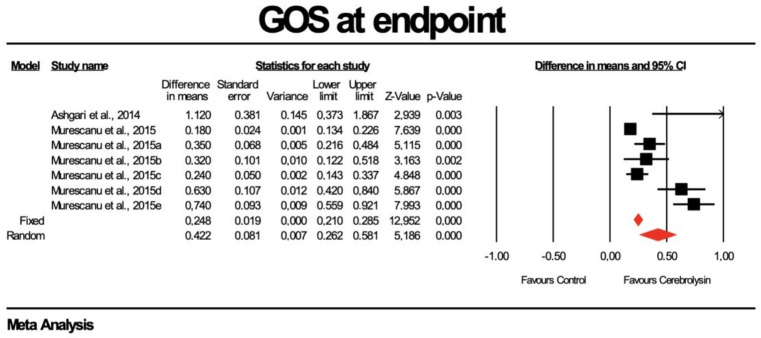
Effect of Cerebrolysin on GOS score (Z value = 12.962, *p* < 0.05). [21,22].

**Figure 4 brainsci-13-00507-f004:**
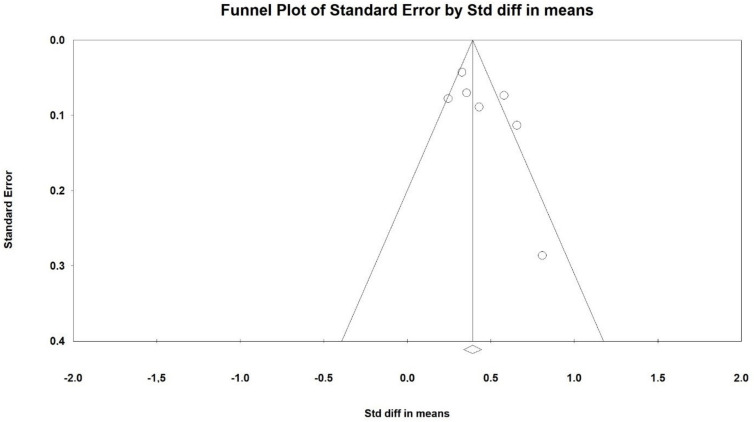
Funnel plot for GOS (DM) in present meta-analysis.

**Figure 5 brainsci-13-00507-f005:**
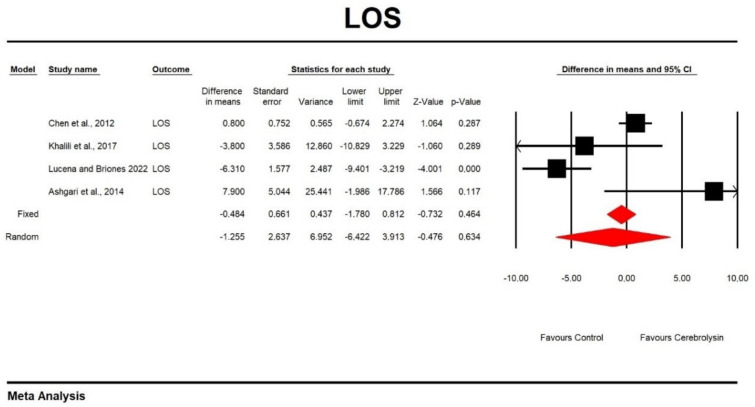
Effect of Cerebrolysin on LOS (Z value = −0.476, *p* = 0.634). [20,21,25,26].

**Figure 6 brainsci-13-00507-f006:**
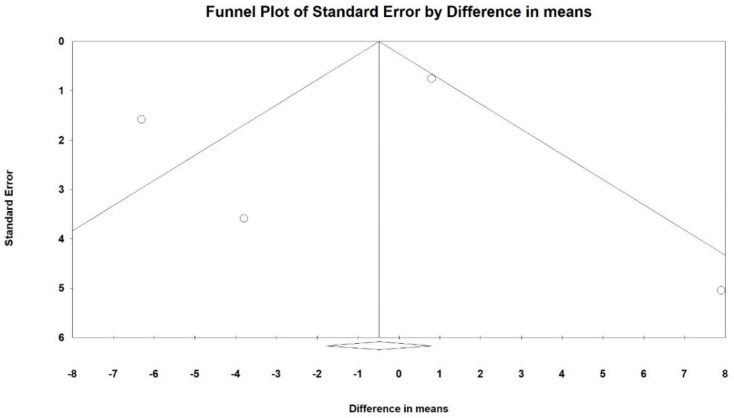
Funnel plot for LOS (DM) in present meta-analysis.

**Figure 7 brainsci-13-00507-f007:**
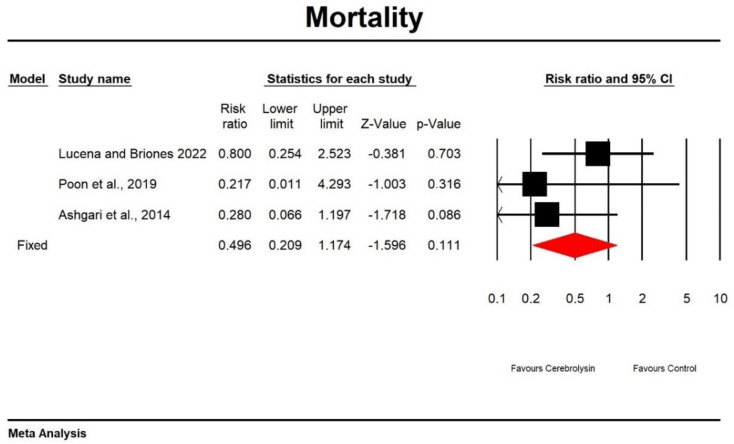
Effect of Cerebrolysin on mortality (Z value = −1.596, *p* = 0.111). [21,26,27].

**Figure 8 brainsci-13-00507-f008:**
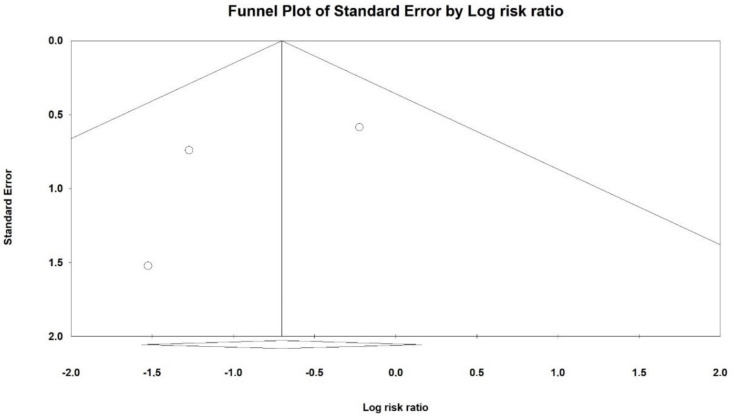
Funnel plot for mortality (RR) in present meta-analysis.

**Table 1 brainsci-13-00507-t001:** Strings used in databases search.

Database	Search Strings with Medical Subject Headings
Pub Med./Cinahl/Web Of Science	(traumatic brain injury OR brain injuries, traumatic OR brain lesion, traumatic OR brain system trauma OR brain trauma OR cerebral trauma OR cerebrovascular trauma OR encephalopathy, traumatic OR mild traumatic brain injury OR organic cerebral trauma OR posttraumatic encephalopathy OR traumatic brain injuries OR traumatic brain injury OR traumatic brain lesion OR traumatic cerebral lesion OR traumatic encephalopathy) AND (cerebrolysin OR cerebrolysin OR cerebrolysine) AND (glasgow outcome scale OR rankin scale OR mini mental state examination OR mortality OR barthel index OR barthel adl index OR barthel index)
Embase	(‘traumatic brain injury’/exp OR ‘brain injuries, traumatic’ OR ‘brain lesion, traumatic’ OR ‘brain system trauma’ OR ‘brain trauma’ OR ‘cerebral trauma’ OR ‘cerebrovascular trauma’ OR ‘encephalopathy, traumatic’ OR ‘mild traumatic brain injury’ OR ‘organic cerebral trauma’ OR ‘posttraumatic encephalopathy’ OR ‘traumatic brain injuries’ OR ‘traumatic brain injury’ OR ‘traumatic brain lesion’ OR ‘traumatic cerebral lesion’ OR ‘traumatic encephalopathy’) AND (‘cerebrolysin’/exp OR ‘cerebrolysin’ OR ‘cerebrolysine’) AND (‘glasgow outcome scale’/exp OR ‘rankin scale’/exp OR ‘mini mental state examination’/exp OR ‘mortality’/exp OR ‘barthel index’/exp OR ‘barthel adl index’ OR ‘barthel index’)

**Table 2 brainsci-13-00507-t002:** Study and patient characteristics. ND—no data, PBO—placebo, NA—not applicable.

Study Characteristics						Intervention	Comparator	Sample Characteristics			
Reference	Country	Sponsorship	Blinding (Y/N)	Trial Duration (Days)	N Total Analyzed	Cerebrolysin Mean Dose/Day (mL); Duration	PBO or Other Intervention	Age (Mean)	N Male	% Male	Seizure (Y/N)
Alvarez et al., 2008 [23]	Spain	industry	N	30 days	59	30 mL/day, 20 infusions over 4 weeks	NA	30.4	40	68	ND
Alvarez et al., 2003 [24]	Spain	industry	N	30 days	20	30 mL/day, 20 infusions over 4 weeks	Y	30.1	15	75	4
Chen et al., 2012 [25]	Taiwan	ND	Y	3 months	32	30 mL/day 5 days	PBO	44.8	21	66	0/32
Khalili et al., 2017 [20]	Iran	academia	N	6 months	129	10 mL/day 30 days	NA	33.3	109	85	Y
Lucena et al., 2022 [26]	Philippines	ND	N	28 days	87	30 mL/day Cerebrolysin for 14 days, 10 mL/day dosage for another 14 days	NA	34	73	84	ND
Poon et al., 2019 [27]	Hong Kong, Taiwan, Republic of Korea, Singapore, Philippines	industry	Y	30 days	40	50 mL of Cerebrolysin daily for 10 days, followed by two additional treatment cycles with 10 mL daily for 10 days	PBO	38.1	32	80	ND
Wong et al., 2005 [14]	China	ND	N	6 months	21	50 mL/day, 20 days	NA	64	13	62	ND
Ashgari et al., 2014 [21]	Iran	ND	N	1 month	53	10 mL/day, 10 days	NA	30	49	92	ND
Murescanu et al., 2015 [22]	Romania	ND	N	1 month	7693	20 mL/day, 10 days	NA	47	5415	70	ND
Murescanu et al., 2015a [22]	Romania	ND	N	1 month	6627	30 mL/day, 10 days	NA	47	5415	70	ND
Muresanu et al., 2020 [15]	Romania	ND	Y	3 months	139	50 mL/day for 10 days, two additional treatment cycles with 10 mL per day for 10 days)	PBO	47.4	123	88.5	ND

**Table 3 brainsci-13-00507-t003:** Surgery qualifications and adverse events in patients treated with Cerebrolysin and Controls.

Reference	Surgery Qualification	Adverse Events			
	Craniotomy (Y/N)	Cerebrolysin Events	Cerebrolysin n	Comparator Events	Comparator n
Alvarez et al., 2008 [23]	ND	ND	ND	ND	ND
Alvarez et al., 2003 [24]	ND	5	20	NA	NA
Chet et al., 2012 [25]	N	ND	32	ND	21
Khalil et al., 2017 [20]	42/36	9	65	4	64
Lucena et al., 2022 [26]	N	ND	42	ND	45
Poon et al., 2019 [27]	ND	0	22	0	24
Wong et al., 2005 [14]	ND	0	21	0	21
Ashgari et al., 2014 [21]	ND	0	25	0	28
Murescanu et al., 2015 [22]	ND	97	1142	541	6151

**Table 4 brainsci-13-00507-t004:** Treatment initiation time, initial GCS, TBI severity in Cerebrolysin and control groups.

Reference	Treatment Initiation Time	Initial GCS	TBI Severity [n]		
			Mild	Moderate	Severe
Alvarez et al., 2008 [23]	23 months	5.5	4	3	32
Alvarez et al., 2003 [24]	23 and 1107 days	6.1	3	1	16
Chet et al., 2012 [25]	24 h	>14	32	0	0
Khalil et al., 2017 [20]	1 month	6.02	*0*	*0*	129
Lucena et al., 2022 [26]	<1 month	5.84	*0*	*0*	87
Poon et al., 2019 [27]	6 h	9.9	0	40	
Ashgari et al., 2014 [21]	48 h	6.75	0	0	53
Murescanu et al., 2015 [22]	48 h	12.72	5125	587	1227

## Data Availability

All the additional data are to be provided after contact with the corresponding author.

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
