# Peer review of "Cerebrolysin in Patients with TBI: Systematic Review and Meta-Analysis"

_brainsci, 2023, doi:10.3390/brainsci13030507_

Round 1

Reviewer 1 Report

The study “Cerebrolysin in patients with TBI: Systematic Review and Meta-Analysis” pooled 10 clinical studies on the effects of cerebrolysin for patients with TBI. Through random effect analysis, the authors confirmed some positive effects but also noted the high heterogeneity in study designs.

However, the high heterogeneity may indicate that pooling the studies for an overall summary is not appropriate, and more information from each individual study is needed, e.g., treatment initiation time, initial outcome measures or measures before treatment, initial severity of TBI, patient age, results from each study.

In the discussion, the authors mentioned four previously performed meta-analysis. However, those articles are not meta-analysis or not about cerebrolysin.

In the conclusion, the authors stated that this study confirmed the safety of Cerebrolysin treatment, but simply listing Table 3 is insufficient.

In addition, the manuscript would benefit from some proofreading to correct typos and check for abbreviations.

A major revision with extensive edits is recommended.

Author Response

Thank you very much for detailed revision. The authors tried to correct the manuscript according to your suggestions. Below we present the answer in point by point response, but we also will send the revised manuscript to the Editors and revised document with color changes here.

Comment 1

"However, the high heterogeneity may indicate that pooling the studies for an overall summary is not appropriate, and more information from each individual study is needed, e.g., treatment initiation time, initial outcome measures or measures before treatment, initial severity of TBI, patient age, results from each study."

Thank you for that comment in particular. 

As we analyzed all manuscripts the severity of TBI, initial GCS, time of drug administration differ in majority.  In additional table 4 after your comment we've placed the data concerning initial GCS (GOS was not available) and severity of TBI and time of drug administration. In primary statistical analysis we tried to extract the data concerning severity of TBI (mild, moderate, severe) and time of administration - but it narrows the subgroups in their number even more.  

Comment 2

"In the discussion, the authors mentioned four previously performed meta-analysis. However, those articles are not meta-analysis or not about cerebrolysin."

Thank you so much for this point- we have corrected the manuscript in order to correct this mistake. 

"To this date, several manuscripts summarizing the effects of Cerebrolysin and other neuroprotective drugs have been published [31,32,33]. One meta analysis involving Cerebrolysin in TBI by Ghaffarpasand et al. in 2018 have been performed."

Comment 3

"In the conclusion, the authors stated that this study confirmed the safety of Cerebrolysin treatment, but simply listing Table 3 is insufficient."

Response:

The mentioned paragraph have been changed. In order to investigate the safety parameters in all those analyzed manuscripts we should perform another additional meta analysis including type of adverse effects and patients concomitant diseases and status during the treatment. There is also heretogenity in most of publications according to what to include as a adverse effect. Concerning all above the manuscript has been changed as presented below:

"Conclusions Our meta-analysis confirms the safety and the positive treatment effect of Cerbrolysin on clincial outcome (GCS and GOS) in patients with TBI. We were not able to detect significant effects on mortality or LOS possibly also due to the relatively small sample size. Therefore, more randomized studies in the field of cerebrolysin in TBI are needed to confirm."

Commet 4

"In addition, the manuscript would benefit from some proofreading to correct typos and check for abbreviations."

Response:

The manuscript has been proofread again to correct typos and abbreviations.

Reviewer 2 Report

The manuscript presented from Konrad Jarosz et al., entitled "Cerebrolysin in patients with TBI: Systematic Review and Meta-Analysis" is interesting and well written. However I would suggest the authors to improve introduction and discussion. The authors mentioned the neuroprotective role of cerebrolysin, however I would suggest the author to improve the text concerning which are the molecular pathways modulated by cerebrolysin in the brain. The authors should motivate the role of this protein. For example: It has been showed that cerebrolysin modulate the expression of different neurotrophic factors, it is possible that the interervent of this factors is relevant and not the administration of cerebrolysin. in parallel other studies showed that cerebrolysin could have a role to alleviate neuroinflammation. The authors should improve this critical point.        

Author Response

Thank you for the detailed and substantive analysis of presented manuscript. The authors tried to respond below by point by point as well as by sending the corrected manuscript to the Editors and revised document with color changes here.

Review:

"The authors mentioned the neuroprotective role of cerebrolysin, however I would suggest the author to improve the text concerning which are the molecular pathways modulated by cerebrolysin in the brain. The authors should motivate the role of this protein. "

"It has been showed that cerebrolysin modulate the expression of different neurotrophic factors, it is possible that the interervent of this factors is relevant and not the administration of cerebrolysin. in parallel other studies showed that cerebrolysin could have a role to alleviate neuroinflammation. The authors should improve this critical point."

Response:

The introduction and discussion have been changed. A section including the molecular mechanisms of Cerebrolyin action has been added, both in terms of mimicking NTF factors and signaling pathways in the brain.

"Cerebrolysin is a low molecular weight neuropeptide preparation obtained from purified porcine brain proteins, that has proven neuroprotective properties in vitro and in vivo, including modeling the permeability of endothelial membranes and anti-inflammatory effects.  Cerebrolysin acts similar to neurotrophic factor (NTF) in order to save the brain tissue from exacerbated inflammatory response. NTF family are small proteins or peptides that includes at least four groups of factors responsible for immunological homeostasis. We distinguish Neurotrophins, Cilliary Neurotrophic Factor subgroup, Glial-cell line-derived neurotrophic factor subgroup, Ephrins and Epidermal Groth Factor and Transforming Growth Factor subgroup. The other mechanism of Cerebrolysin neuroprotection involves sonic hedgehog (SHH) pathways. Shh plays a significant role in embryonic but also adult neuronal damage signaling. Recent TBI models showed that Shh pathway is exacerbated after cortical injury. Cerebrolysin modulates mRNA expression in order to activate Shh pathway itself but also by Shh receptors activation (gliogenesis, neurogenesis). The effector fot Shh pathways signaling is Gli protein complex. Gli complex is involved not only in developmental processes but also neurorecovery in pathological conditions. Acting on neurotropic agents Cerebrolysin changes the activity of  GABAergic and cholinergic pathways in brain. Cerebrolysin also has been proven to affect particularly on microglia cells. Those glia cells control the homeostasis and immunological response and Cerebrolysin modulate their function in order to limit the exacerbated  inflammation response.  

In vivo studies with rats have shown the reduction of astrocyte activation, reduction in axonal damage and increasing in neurogenesis in closed experimental brain injury. In vivo studies have with patients also shown its beneficial role in slowing down of EEG activity and improving rehabilitation results. 

According to the recommendation Cerebrolysin could be administrated in various dosages 10-50ml per day. The intravenous infusion should be prepared in 100ml total volume after adding the Cerebrolysin to 0,9% saline or  Ringers solution or 5% glucose solution. The duration of infusion should take at least 15 minutes. 

Cerebrolysin in contraindicated in patients with allergy to Cerebrolysin or any of its compounds in history and also in patients with history of seizure, or  severe renal failure.

Round 2

Reviewer 1 Report

The revision addressed all of my comments. Please edit the format of Table 4 to match with other tables. I recommend the manuscript for publication.

Reviewer 2 Report

The authors improved the manuscript